# Characterization of Traditional Chinese Sesame Oil by Using Headspace Solid-Phase Microextraction/Gas Chromatography–Mass Spectrometry, Electronic Nose, Sensory Evaluation, and RapidOxy

**DOI:** 10.3390/foods11223555

**Published:** 2022-11-08

**Authors:** Yan Chen, Yingjie Fu, Peng Li, Hui Xi, Wuduo Zhao, Dingzhong Wang, Jian Mao, Shusheng Zhang, Shihao Sun, Jianping Xie

**Affiliations:** 1Flavor Research Center, Zhengzhou University, Zhengzhou 450001, China; 2The Key Laboratory of Tobacco Flavor Basic Research of CNTC, Zhengzhou Tobacco Research Institute, Zhengzhou 450001, China; 3Center of Advanced Analysis and Gene Sequencing, Zhengzhou University, Zhengzhou 450001, China

**Keywords:** sesame oil, HS-SPME/GC-MS, odor activity value, electronic nose, oxidative stability

## Abstract

Xiao Mo Xiang You (XMXY) is a traditional Chinese sesame oil variety that is obtained through a hot water flotation process. This unique process gives the oil a unique aroma, health benefits, and excellent product stability. Although XMXY is always the most expensive among all the sesame oil varieties, it is usually used as a flavoring in many traditional Chinese daily food products and is increasingly popular. In order to reveal the characteristics of the oil, the volatile components, sensory evaluation, and oxidation stability of five XMXY samples were, respectively, analyzed by using headspace solid-phase microextraction/gas chromatography–mass spectrometry, an electronic nose, sensory evaluation, and RapidOxy. Comparisons and multidimensional statistical analysis were also carried out to distinguish XMXY from roasted sesame oil (RSO) and cold-pressed sesame oil (CSO) samples. In total, 69 volatiles were identified from XMXY, RSO, and CSO samples. Some compounds possessed high odor activity value (OAV > 1) in XMXY, including heterocyclic compounds, phenols, and sulfur-containing compounds. Additionally, they were also the main volatile components that distinguish XMXY from RSO and CSO. Roasted and nutty aromas were the dominant aroma attributes of XMXY. XMXY had better flavor intensity and oxidation stability than the other two sesame oil samples. These results are very valuable for the quality control and product identification of traditional Chinese sesame oil.

## 1. Introduction

Sesame (*Sesamum indicum* L.) is considered an important oilseed crop that is widely cultivated in Asia, Africa, and Central and South America [1]. Sesame seeds can be processed in various ways to obtain sesame oil. Sesame oil contains many kinds of natural phytochemicals, such as sesamin, sesamol, and sesamolin, which are known to possess anti-inflammatory and anticancer activities [2,3,4,5]. Sesame oil varieties are usually divided into refined, roasted, and cold-pressed oil according to their processing methods [6]. The aroma of the sesame oil is a characteristic indicator for consumers to select it. Most aroma compounds of the oil are formed by various reactions during processing, for example, enzymatic reactions and thermal reactions such as Maillard reaction, Strecker degradation, caramelization, and lipid thermal reaction [7,8]. Many studies also showed that the roasting temperature of sesame seeds has a significant impact on the concentration of volatiles, and pyrazines are major volatile compounds in sesame oil [9,10,11]. As a result, the aroma of sesame oil is very different due to different processing techniques. In addition, sesame oil varieties are also obviously different in the aspect of product stability for the same reasons [12,13,14].

Hot water flotation is a traditional processing method to produce sesame oil in northern China, through which the oil is obtained from roasted sesame seeds and does not undergo refining; in this way, many distinct sensory odors are preserved [6]. The sesame oil obtained with this technological processing is named “Xiao Mo Xiang You” (XMXY) in China. This unique process gives the oil a unique aroma, health benefits, and excellent product stability. Although this variety is always the most expensive of all types of sesame oil, it is usually used as a flavoring in the traditional diet of the Huanghuai region in China and is increasingly popular [15].

The aroma of sesame oil can largely reflect its quality, and consumers often choose sesame oil varieties with better sensory quality as their daily condiment oil [16]. Over the last ten years, many studies have concentrated on the volatile compounds in different sesame oil products with different processing conditions [16,17]. The volatile profiles of various sesame oil products are usually analyzed using gas chromatography–mass spectrometry (GC-MS) coupling with headspace solid-phase microextraction (HS-SPME) [18,19]. Hundreds of volatile compounds have been identified in various sesame oil types, including esters, alcohols, acids, ketones, heterocyclic compounds, etc. [20,21,22]. Studies have also shown that the ingestion of oxidized oil can lead to a variety of human diseases, such as cancer, atherosclerosis, and Alzheimer’s disease [23,24,25,26]. However, studies on the volatile characterization and oxidative stability of the sesame oil produced via hot water flotation are limited.

The purpose of this study was to reveal the characterization of XMXY. The volatile components of XMXY, roasted sesame oil (RSO) and cold-pressed sesame oil (CSO) samples were analyzed by using HS-SPME/GC-MS and an electronic nose. Based on the qualitative and quantitative data of the volatile components, the differences in XMXY from different sesame oil varieties were compared through multivariate statistical analysis. Sensory evaluation was also conducted to show the unique organoleptic property of XMXY. Moreover, the oxidation stability of sesame oil samples was evaluated through RapidOxy under accelerated conditions.

## 2. Materials and Methods

### 2.1. Chemicals and Materials

Five sesame oil samples obtained through hot water flotation (XMXY1 to XMXY5), an RSO sample, and a CSO sample were purchased from Zhoukou county, Henan province, China. All sesame oil samples were stored in sealed brown glass containers at 4 °C until further analysis. 2-Methyl-3-heptanone was used as an internal standard, and n-alkane mixtures (C7-C30) were purchased from Merck (Darmstadt, Germany). SPME fibers (50/30 μm DVB/CAR/PDMS; 100 μm PDMS; 85 μm PA) were purchased from Supelco Co. (Bellefonte, PA, USA).

### 2.2. HS-SMPE/GC-MS Analysis of Sesame Oil Samples

The volatile components of sesame oil samples were analyzed using a reported HS-SMPE/GC-MS method [18], with slight modifications. A CAR/DVB/PDMS fiber was employed for the SPME of the samples, and the fiber was preconditioned for 30 min at 270 °C before the extraction of samples. Briefly, 1 g of each sesame oil sample was transferred into a 4 mL headspace sample vial. The vial was equilibrated at 60 °C for 20 min in a thermostatic water bath after tightening the cap. Then, the aged DVB/CAR/PDMS fiber head was inserted into the headspace of the sample vial. After 60 min, the fiber was instantly injected into the GC-MS injection port, which desorbed for 5 min at 250 °C. 

GC-MS analysis was performed using an Agilent 7890B gas chromatograph coupled with a 5977B series mass spectrometer (Santa Clara, CA, USA). The volatile components were, respectively, separated in the column systems of two different polarities: HP-5MS (60 m × 250 μm × 0.25 μm) and DB-WAXETR (60 m × 250 μm × 0.25 μm). The operating conditions were modified appropriately according to the method of Jia et al. [18]. For the HP-5MS column, the oven temperature was set at 40 °C, increased to 160 °C with a temperature ramp of 2 °C/min, and finally increased to 240 °C with a temperature ramp of 10 °C/min. For the DB-WAXETR column, the oven temperature at 40 °C was increased to 170 °C with a temperature ramp of 2.5 °C/min, maintained for 3 min, and finally increased to 240 °C with a temperature ramp of 10 °C/min and maintained for 4 min. A splitless mode was used, and a flow rate of 1.0 mL/min was helium as the carrier gas. The temperature of the ion source was 250 °C, the ionization energy was 70 eV, and the scan mass range was 35–350 aum.

### 2.3. Identification, Quantification, and Odor Activity Value (OAV) Calculation of the Volatile Components

The tentative identification of the volatiles in sesame oil samples was based on the mass spectra in the NIST 17.0 database. The retention indices (RIs) were obtained to further confirm each compound by comparing its values with those reported in the literature [27]. The RIs of the compounds were calculated according to the formula:(1)$$\mathrm{RI}=100n+100\times \frac{t_{Ru}-t_{Rn}}{t_{R\left(n+1\right)}-t_{Rn}}$$
where *n* is the number of carbon atoms in a series of normal alkanes, and *t_Ru_* is the retention time of the volatile compound. *t_Rn_* and *t_R_*_(*n*+1)_ are the retention times of the normal alkanes.

The volatile compounds in sesame oil samples were quantified by using a semi-quantitative method [28]. The relative concentration of volatile compounds was obtained by comparing the peak areas of the compounds and the internal standard (2-methyl-3-heptanone).

The OAVs of all volatile compounds were calculated according to the following formula:(2)$$\mathrm{OAVs}=\frac{\mathrm{Ci}}{\mathrm{OTi}}$$
where Ci is the relative concentration of the volatile compound calculated by using semi-quantitation. OTi is its corresponding order threshold by adopting the method of Tian et al. [29].

### 2.4. Electronic Nose Analysis

An electronic nose (PEN 3, Airsense Analytics Co., Ltd., Schwerin, Germany) was used to analyze the sensory profile of the sesame oil samples, which had ten aroma receptors. These sensors were named W1C, W5S, W3C, W6S, W5C, W1S, W1W, W2S, W2W, and W3S, were, respectively, sensitive to substances such as aromatic hydrocarbons, nitrogen oxides, aromatic ammonia, hydrogen compounds, short-chain alkane aromatic compounds, methyl-containing compounds, sulfides, alcohols, and carbonyl compounds, organic sulfides and aromatic components, long chain alkanes. Briefly, 5 g of sesame oil was transferred into a 25 mL headspace bottle, covered and sealed, and allowed to stand at 26 °C for 30 min before testing. In terms of electronic nose measurement parameters, cleaning time was 120 s; zero return time was 5 s; pre-injection time was 8 s; measurement time was 60 s, and carrier gas flow rate was 400 mL/min.

### 2.5. Sensory Analysis 

Sensory analysis was performed by ten panelists (five males and five females, aged 20–30) from the Flavor Research Center at Zhengzhou University, China. The panelists were recruited based on their availability and experience, and they were trained at least three times according to ISO 8586. The quantitative descriptive analysis (QDA) was carried out in standard sensory booths with temperature control (22–24 °C) and air circulation. The sesame oil samples stored in the refrigerator were taken out and kept at room temperature for subsequent sensory evaluation. Each sample (5.0 mL) was loaded into glass bottles (10.0 mL) coded with 3-digit numbers and then presented to the panelists randomly. To evaluate the overall flavor profile of sesame oil samples, the panelists discussed and selected six sensory attributes of sesame oil samples, including roasted, nutty, sweet, smoky, pungent, and green. A 5-point scale ranging from 0 to 5 was applied to quantitatively evaluate the contribution of each sensory attribute to the overall flavor profile of the sesame oil. Additionally, 0 meant no odor, and 5 meant extremely strong odor. The results were expressed as the average score of 3 times in the spider diagram.

### 2.6. Oxidation Stability Analysis

The oxidation stability analysis of sesame oil samples was evaluated using a RapidOxy reactor (Anton Paar, Blankenfelde-Mahlow, Germany). A previously reported method was adopted [30], with some modifications. In each experiment, 3 g of one of the sesame oil samples was accurately weighed under the conditions of a temperature of 150 °C and pressure of 700 kpa, and the time of reaching a pressure drop of 10% was measured, which is the induction period (IP). The test of each sample was replicated six times.

### 2.7. Statistical Analysis

SPSS 26.0 (IBM, New York, NY, USA) was used to analyze the data with a one-way ANOVA (confidence level *p* < 0.05). SIMCA 14.1 (MKS Umetrics, Umea, Sweden) was used for the orthogonal partial least-square discriminant analysis (OPLS-DA). The radar chart was plotted using Origin 2018 (Origin Lab Corporation, Northampton, MA, USA). The heat maps of the samples were drawn by using TBtools (Toolbox for Biologists; version 1.082, Guangzhou, China).

## 3. Results and Discussion

### 3.1. HS-SPME/GC-MS Analysis of Sesame Oil Samples

Firstly, some main parameters of HS-SPME were optimized according to the total peak areas, such as the tape of SPME fiber, extraction, and desorption conditions. The results (Appendix A) showed that a CAR/PDMS/DVB fiber, extraction at 60 °C for 60 min, and desorption at 250 °C for 5 min were suitable experimental conditions for the HS-SPME analysis of volatile components in sesame oil. In order to study the unique flavor of XMXY, in addition to the five XMXY samples, we also analyzed RSO and CSO samples. 

As shown in Appendix A and Figure 1, 69 compounds were identified from the seven samples by comparing their RI and mass spectrum information, with the compounds referenced in the NIST17 mass spectra, comprising 22 N-heterocyclic compounds, 9 O-heterocyclic compounds, 9 S-containing volatile compounds, 8 phenols, 7 acids, 4 ketones, 4 aldehydes, 4 alcohols, 1 ester, and 1 alkene. The heterocyclic compound was the most abundant group in XMXY and RSO samples, while the main compounds in CSO were acids and aldehydes. More kinds of volatile components were identified from the sesame oil obtained by roasting sesame seeds, which was consistent with previous studies [8,9,10,18].

The content of N-heterocyclic compounds (118.29–191.91 μg/g) was the highest in the volatile components of all the XMXY samples. Pyrazines mainly existed in the form of alkylated pyrazines, and generally had a strong aroma when roasted and nutty [31]. Additionally, pyrazines were mostly derived from the Strecker degradation pathway of the seeds during high-temperature roasting according to previous studies [7,32]. No pyrazine compound was found in the CSO sample, the reason for which may be that CSO was produced at lower temperatures.

Nine O-heterocyclic compounds were found in the XMXY samples, mainly including furans, furanones, and their derivatives. The total content of O-heterocyclic compounds (32.00–68.26 μg/g) was relatively high in the volatile components of the XMXY samples. O-heterocyclic compounds are usually produced through the caramelization reaction that occurs during the roasting of oil seeds [8]. The thresholds of the compounds were low, giving the XMXY sample its sweet, nutty, fruity, and caramel-like characteristics [33].

Eleven S-containing compounds were identified in this study, mainly including thioethers, thiols, and others. Dimethyl disulfide (found in onion, cabbage, etc.) may be produced by the Stray degradable reaction, especially in some vegetable oil varieties rich in sulfur-containing amino acids [34]. Methanethiol (putrid, cabbage-like, sulfury) was identified in almost all of the XMXY samples, which had a low threshold and played an important role in the flavor profile of the XMXY samples [35].

Eight phenols were identified in the volatile compounds of the XMXY samples. 2-Methoxy-phenol (14.84–85.49 μg/g) and 2-Methoxy-4-vinylphenol (4.08–23.78 μg/g) were the components with higher content in the XMXY samples. Phenols may play an important role in the formation of the XMXY aroma profile and are known to significantly contribute to the smoky note because of their low odor thresholds [36].

Other compounds were found in low concentrations in the volatile compounds of XMXY, but they were the main components of cold-pressed sesame oil, namely short-chain aldehydes and pelargonic acid, which gave it an associative and fruity aroma [37].

Orthogonal partial least-squares discriminant analysis (OPLS-DA) is a supervised statistical method of discriminant analysis, which is different from the principal component analysis [38]. In order to explore the aroma compounds profile of XMXY, an orthogonal partial least-squares discriminant analysis (OPLS-DA) model (Figure 2A) was established based on the volatile compounds in XMXY and RSO samples. A loading S-plot and variable importance in the projection (VIP) plot (Figure 2B,C) were also adopted to confirm the key aroma compounds in XMXY. As shown in Figure 2C, variables with a VIP value greater than 1 (*p* < 0.05) were considered the volatile compounds that caused a significant difference [39]. The variables with significant differences mainly included heterocyclic compounds, phenols, and sulfur-containing compounds.

### 3.2. Identification of Aroma Active Compounds by OAVs

Among the volatile components of vegetable oil, only a small number of components directly affect the organoleptic properties of vegetable oil. Abundant studies showed that the concentration and odor threshold of volatile compounds were the key factors that determined the contribution of volatile compounds to the overall aroma characteristics [8].

OAV is the ratio of the concentration to the odor threshold of oil. It was possible to compare the contribution of each compound to the aroma characteristics in the sesame oil samples by the calculation of OAV [40]. The volatile compounds with an OAV greater than 1 were considered the key aroma-active compounds [41]. As shown in Table 1, the OAVs of the volatile components in XMXY were generally higher than that of the RSO and CSO. The volatile components with larger OAVs in XMXY mainly included methanethiol, dimethyl disulfide, methyl-pyrazine, 2-ethyl-6-methyl-pyrazine, 3-ethyl-2,5-dimethyl-pyrazine, 2-furanmethanol, 2-methoxy-4-vinylphenol, 2-methoxy-phenol, etc. These compounds were important contributors to the aroma profile of XMXY samples. In addition, N-heterocyclic compounds, S-containing compounds, and phenols were also the components with high OAVs in RSO. Hexanal and hexanoic acid were relatively high-value compounds in CSO but were not found in XMXY and RSO. 

### 3.3. Electronic Nose Analysis of Sesame oil Samples

An electronic nose can detect simple or complex smells by simulating the human smelling process [42]. With the advantages of high speed and low cost, electronic noses have been widely used to study the volatile components of many agricultural products [43]. In this study, an electronic nose was used to analyze the volatile components of XMXY, RSO, and CSO samples. The radar chart drawn from the electronic nose data is shown in Figure 3A. The ten sensors of the electronic nose had different responses to the volatile components in the sesame oil samples, which was similar to the results reported by Hai et al. [44]. The aroma profile of the RSO sample was similar to that of the XMXY samples, but the overall aroma intensity of almost all XMXY samples was stronger than that of RSO. The response intensity of the volatile compounds in CSO was weak on all sensors. In order to evaluate the flavor difference between XMXY and the other two sesame oil samples, XMXY3 with the most intense flavor was used as the representative, and the results are shown in Figure 3B. There were significant differences (*p* < 0.05) between XMXY3 and CSO on all 10 sensors. Except for the two sensors W1C and W3S, the responses of XMXY3 and RSO samples to the other eight sensors were significantly different (*p* < 0.05). In addition, the response of XMXY3 was significantly higher on W1W, W2W, and W5S sensors, indicating that XMXY3 may contain a higher concentration of organic sulfides and nitrogen oxides. The results of the electronic nose were consistent with those obtained through GC-MS analysis.

### 3.4. Sensory Analysis of Sesame Oil Samples

Six odor attributes were chosen for the evaluation of sesame oil samples, and the results are shown in Figure 4, which reveal that roasted and nutty were the dominant aroma attributes of XMXY, followed by smoky, pungent, sweet, and green. The evaluation of RSO was similar to XMXY, but its aroma intensity was lower than that of XMXY, and the roasted and nutty aromas of XMXY were more intense. Furthermore, a one-way ANOVA was used to distinguish the differences (*p* < 0.05) in sensory evaluation scores, and it was found that the aroma profile of CSO was significantly different from that of XMXY and RSO, green was the dominant aroma attributor, and the intensity of other aroma characteristics was weak. The sensory analysis results were basically consistent with the results obtained through gas chromatography analysis and the use of E-nose, further confirming the accuracy of the previous experimental results.

### 3.5. Oxidation Stability Analysis of Sesame Oil Samples

In this study, the oxidation stability of the sesame oil samples was evaluated under accelerated oxidation conditions by using a RapidOxy reactor. The results are shown in Table 2, which reflects the induction time of XMXY1–5, RSO, and CSO samples under different pressure drops. Compared with CSO (32.11 min), XMXY and RSO samples had longer induction time (48.43–61.52 min) and better oxidative stability. CSO is produced from unheated sesame seeds, whereas antioxidant active components such as sesamol and Maillard reaction products are produced during sesame roasting, which may be the reason for the higher oxidative stability of XMXY and RSO samples [45,46,47]. This conclusion could be verified in the analysis of GC-MS.

## 4. Conclusions

In this study, traditional Chinese sesame oil (XMXY) was analyzed using HS-SPME/GC–MS, an electronic nose, sensory evaluation, and RapidOxy. In total, 69 volatile compounds were identified from XMXY, RSO, and CSO samples. In terms of the identified volatile components, multidimensional statistical analysis and OAV value analysis revealed that heterocyclic compounds, phenols, and sulfur-containing compounds were the key volatile compounds in XMXY and contributed more to the overall aroma. Roasted and nutty aromas were the dominant aroma attributes of XMXY. The XMXY sample obtained through hot water flotation had better flavor and oxidation stability than the other two sesame oil (RSO and CSO) samples. This study provides some information on the quality control of traditional sesame oil. Future research will focus on the formation mechanism of the traditional sesame oil flavor, the precise control of product quality, etc.

## Figures and Tables

**Figure 1 foods-11-03555-f001:**
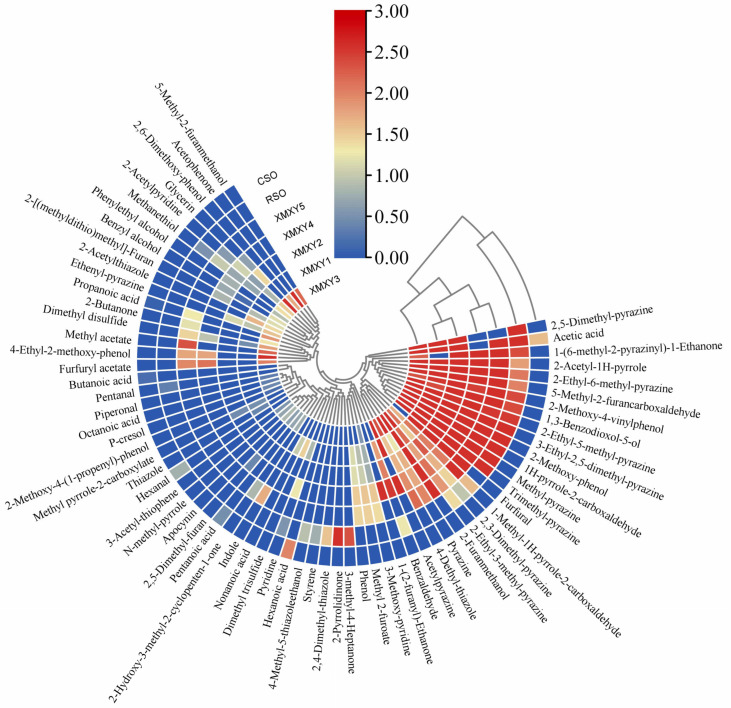
A heat map of volatile compounds in XMXY1–5, RSO, and CSO samples. XMXY1–5: Xiao Mo Xiang You 1–5; RSO: roasted sesame oil; CSO: cold-pressed sesame oil.

**Figure 2 foods-11-03555-f002:**
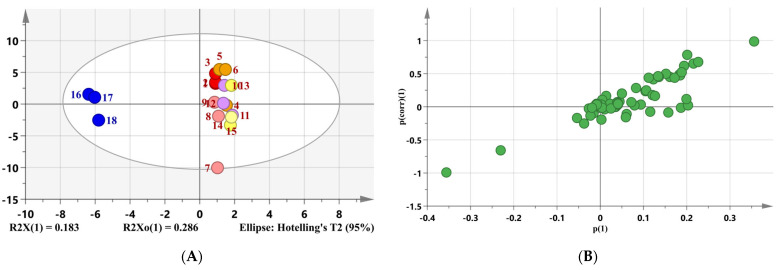
OPLS-DA of XMXY and RSO samples: (**A**) the score plot, 1–3: Xiao Mo Xiang You 1; 4–6: Xiao Mo Xiang You 2; 7–9: Xiao Mo Xiang You 3; 10–12: Xiao Mo Xiang You 4; 13–15: Xiao Mo Xiang You 5; 16–18: roasted sesame oil; (**B**) the loading S-plot; (**C**) the VIP plot.

**Figure 3 foods-11-03555-f003:**
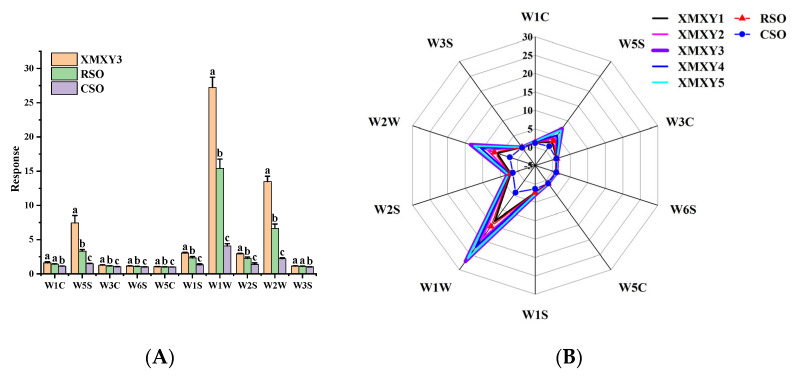
(**A**) Radar chart, different superscripts indicate statistically significant different values (*p* < 0.05); (**B**) analysis of the E-nose response data. XMXY1–5: Xiao Mo Xiang You 1–5; RSO: roasted sesame oil; CSO: cold-pressed sesame oil.

**Figure 4 foods-11-03555-f004:**
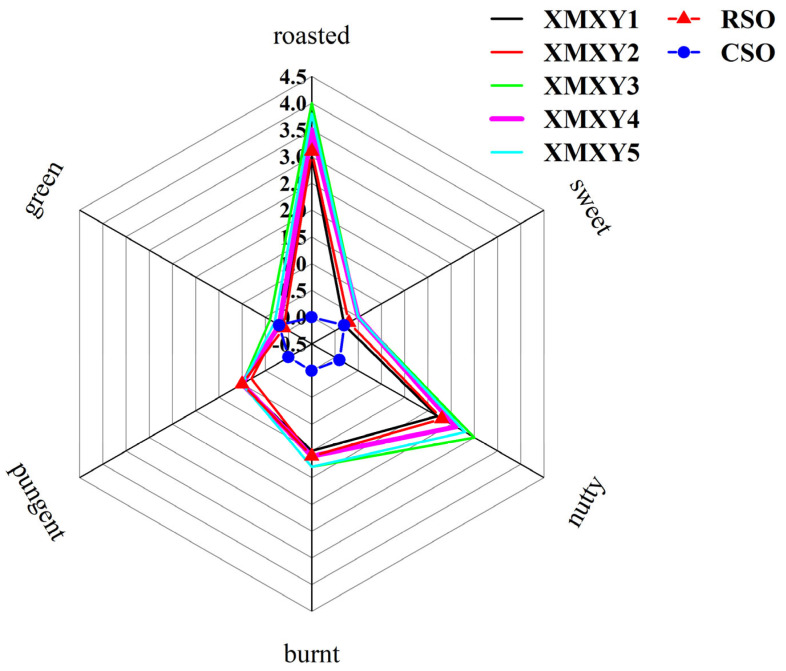
Aroma profile of XMXY1–5, RSO, and CSO samples. XMXY1–5: Xiao Mo Xiang You 1–5; RSO: roasted sesame oil; CSO: cold-pressed sesame oil.

**Table 1 foods-11-03555-t001:** Odor thresholds and OAVs of the compounds in XMXY1–5, RSO, and CSO samples.

Name	CAS ID	OTi ^a^ (mg/kg)	OAVs ^b^						
			XMXY1	XMXY2	XMXY3	XMXY4	XMXY5	RSO	CSO
Methanethiol	74-93-1	0.00006	--	10,333	26,333	10,666	17,000	7166	--
Methyl acetate	79-20-9	2	--	--	2	--	2	--	--
Pentanal	110-62-3	0.24	--	--	--	--	--	1	--
Dimethyl disulfide	624-92-0	0.012	32	--	208	77	145	--	--
Hexanal	66-25-1	0.12	--	--	--	--	--	--	16
Thiazole	288-47-1	0.038	8	--	--	--	--	--	--
Methyl-pyrazine	109-08-0	0.06	613	639	1031	688	717	475	--
4-Dethyl-thiazole	693-95-8	0.055	35	39	75	51	64	--	--
2,4-Dimethyl-thiazole	541-58-2	0.018	--	--	--	--	--	107	--
2,5-Dimethyl-pyrazine	123-32-0	2.6	6	5	10	--	--	7	--
2,3-Dimethyl-pyrazine	5910-89-4	0.4	4	9	8	5	4	2	--
Dimethyl trisulfide	3658-80-8	0.0025	712	--	--	--	--	--	--
2-Ethyl-6-methyl-pyrazine	13925-03-6	0.04	168	146	366	196	158	80	--
2-Ethyl-5-methyl-pyrazine	13360-64-0	0.32	22	15	29	17	14	14	--
2-Ethyl-3-methyl-pyrazine	15707-23-0	0.13	26	18	44	25	20	12	--
Trimethyl-pyrazine	14667-55-1	0.29	23	22	44	23	19	23	--
Ethenyl-pyrazine	4177-16-6	0.7	1	1	3	--	--	--	--
Acetic acid	64-19-7	0.124	57	60	49	--	183	247	5
3-Ethyl-2,5-dimethyl-pyrazine	13360-65-1	0.024	487	424	815	419	431	365	--
Furfural	98-01-1	0.77	20	22	--	15	25	--	--
Benzaldehyde	100-52-7	0.32	7	9	--	14	--	4	--
Propanoic acid	79-09-4	0.72	--	--	4	--	2	--	--
5-Methyl-2-furancarboxaldehyde	620-02-0	0.5	33	36	10	28	49	6	--
2-Acetylpyridine	1122-62-9	0.019	41	29	97	58	27	--	--
Acetylpyrazine	22047-25-2	0.06	202	--	109	150	--	--	--
2-Acetylthiazole	24295-03-2	0.01	216	--	168	--	--	--	--
Acetophenone	98-86-2	5.629	--	--	1	--	--	--	--
2-Furanmethanol	98-00-0	0.0123	1086	1302	1513	1315	1560	--	--
1-(6-methyl-2-pyrazinyl)-1-Ethanone	22047-26-3	0.3	0.3	--	19	12	23	33	14
2-[(methyldithio)methyl]-Furan	57500-00-2	0.00004	21499	--	31250	19500	--	--	--
Pentanoic acid	109-52-4	0.061	--	--	10	--	--	--	6
Hexanoic acid	142-62-1	0.7	--	--	--	--	--	--	4
2-Methoxy-phenol	32994	0.016	927	1772	5343	3423	2826	347	--
Phenol	108-95-2	0.1	12	12	--	18	17	--	--
4-Ethyl-2-methoxy-phenol	2785-89-9	0.05	--	--	261	48	47	--	--
P-cresol	106-44-5	0.025	--	--	14	--	--	--	--
1-Methyl-1H-pyrrole-2-carboxaldehyde	1192-58-1	0.37	0.37	8	8	6	8	16	5
2-Methoxy-4-vinylphenol	7786-61-0	0.2	95	104	82	90	475	85	--
2,6-Dimethoxy-phenol	91-10-1	0.05	--	--	9	--	--	--	--
2-Methoxy-4-(1-propenyl)-phenol	97-54-1	0.263	0.1	--	--	1	--	--	--
Indole	120-72-9	0.1	--	--	--	--	7	--	--
Apocynin	498-02-2	0.3	--	--	1	--	--	--	--

Abbreviations: OTi ^a^ was referenced from the book *Odor Thresholds: Compilations of Odor Threshold Values in Air, Water and Other Media* (second enlarged and revised edition) (van Gemert, 2011); OAVs ^b^ refer to the odor activity values; XMXY1–5: Xiao Mo Xiang You 1–5; RSO: roasted sesame oil; CSO: cold-pressed sesame oil.

**Table 2 foods-11-03555-t002:** Induction time of XMXY1–5, RSO, and CSO samples.

Pressure Drop (%)	Induction Time						
	XMXY1	XMXY2	XMXY3	XMXY4	XMXY5	RSO	CSO
2%	42.17 ± 0.68	39.48 ± 1.22	53.11 ± 1.11	50.44 ± 1.54	48.83 ± 1.13	42.74 ± 1.08	28.8 ± 1.96
4%	45.59 ± 0.93	41.74 ± 1.44	58.18 ± 0.85	55.77 ± 1.59	54.18 ± 1.4	46.78 ± 1.09	29.76 ± 2.01
6%	46.73 ± 0.92	42.64 ± 1.36	59.64 ± 0.73	57.25 ± 1.53	55.63 ± 1.53	49.08 ± 1.19	30.58 ± 2.04
8%	47.61 ± 0.92	43.44 ± 1.31	60.59 ± 0.8	58.21 ± 1.55	56.49 ± 1.49	50.85 ± 1.38	31.34 ± 2.06
10%	48.43 ± 0.92	44.21 ± 1.27	61.52 ± 0.82	59.09 ± 1.55	57.34 ± 1.46	52.27 ± 1.47	32.11 ± 2.07

Abbreviations: XMXY1–5: Xiao Mo Xiang You 1–5; RSO: roasted sesame oil; CSO: cold-pressed sesame oil.

## Data Availability

Data are contained within the article or Appendix A.

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
