# Peer review of "Characterization of Traditional Chinese Sesame Oil by Using Headspace Solid-Phase Microextraction/Gas Chromatography–Mass Spectrometry, Electronic Nose, Sensory Evaluation, and RapidOxy"

_foods, 2022, doi:10.3390/foods11223555_

Round 1
Reviewer 1 Report
The paper “Characterization of traditional Chinese sesame oil by head- space solid phase microextration/gas chromatography-mass spectrometry, electronic nose, sensory evaluation and RapidOxy” is focused on determining the volatile components, sensory perception, and oxidation stability of different samples of sesame oil. The sesame oil samples were obtained through different techniques: hot water flotation, roasted sesame oil, and cold pressed oil.
The experimental design is well chosen. The analytical methods used in the study were headspace solid phase microextration/gas chromatography-mass spectrometry, electronic nose, sensory evaluation, and RapidOxy. This methodology along with the statistical methods allowed an extensive characterization of the oil samples. Statistical analysis is relevant for the analytical methods used in the study.
Some minor observations are:
Lines 68-69 “Moreover, comprehensive comparisons were 68 also carried out to reveal its characterization different from other sesame oils” –my suggestion is to rephrase this sentence because is not enough clear. What kind of “other oils”?
Please check carefully and specify the producer, city, and country for all the analytical equipment presented in the Materials and Methods section; in some cases, this information is missing
Sensory analysis section- the panelists were trained or untrained? Please specify.
Line 141- The authors need to complete the sentence “The results were consistent with previous studies” with more citations
The results of electronic nose analysis need to be compared with results obtained by other authors who used this method
Reviewer 2 Report
The manuscript:"Characterization of traditional Chinese sesame oil by headspace solid phase microextration/gas chromatography-mass spectrometry, electronic nose, sensory evaluation and RapidOxy" is a very interesting study on the evaluation of sesame oil samples with different preparation methods. Manuscript is well written, but lacks some data necessary to better understand the relationship between the obtained results. A detailed review of the manuscript is provided below.
Abstract:
Lines 13-18: in this part of the abstract, please use the present tense
Line 17: instead of sensory perception it is better to use the phrase: sensory evaluation
In the abstract, there is no data on the XMXY 1 to 5 samples, at the same time the results obtained for these samples can be found in the further part of the work.
There is no summary of the research in the abstract in terms of explaining the practical significance of the conducted research, what it gave, what comments can be made to the producers of the tested oil on the basis of obtained results?
Introduction
There should be spaces before the parentheses of the quoted literature. This applies to all of manuscript.
In the introduction, please describe the nutritional and health properties of sesame oil.
In this part of the manuscript please explain in more detail the purpose of the undertaken research.
Line 45: "hot water flotation" -> what is the water temperature of flotation?
Line 53: does the aroma of sesame oils reflect their quality in a negative or positive way? Please explain.
Materials and Methods
Materials
Please explain what samples were tested. What was the number of samples of each type tested?
Please provide details of the method of obtaining individual samples (temperature and time of flotation, roasting and cold press).
What are the differences between the samples XMXY1 to XMXY5?
Methods
Line 98: please explain what the abbreviation: OAVs means.
Sensory analysis
Please describe in more detail the method of the sensory evaluation carried out, refer to the literature, i.e. ISO standards. What method was used? How many members did the team of experts consist of (there are two numbers: 7 and 10 in manuscript)? Were the experts trained? According to what standard? Under what conditions were the evaluations carried out? Were the tested samples specially prepared for evaluation?
Statistical analysis
Please add which test was used to examine statistically significant differences between individual samples. What level of significance was used?
Results and discussion
The figures shown are illegible. Please improve their quality.
Under the tables and figures there is no explanation of the entire name of the samples tested. Please add them.
Line 155: Table S1? What does it mean? Why was the analysis carried out on 7 samples, which samples?
Line 196: what does abbreviation VIP mean? Please explain.
Line 204: what does abbreviation OAVs mean? Please explain.
Table 1: please explain in more detail what abbreviation OTs stands for. An explanation of the abbreviations introduced below the table would facilitate the analysis of the obtained data contained in table 1. Were the analyzes, the results of which are presented in Table 1, performed in duplicate? can the obtained data be subjected to statistical analysis?
Lines 236-237: The results of electronic nose were not consistent with GC-MS analysis. Because, if I interpret the obtained results correctly, nitrogen oxides were not determined in GC-MS analysis. Please check.
Were the obtained results of the sensory analysis statistically significant? please include statistical analysis in manuscript.
Do the compounds marked at work show toxic properties, and if so, which ones? What levels of these compounds are safe for consumers?
Conclusion
This section should contain the conclusions that were obtained from the carried out research. Was the applied flotation beneficial from a nutritional point of view? Or maybe roastong would be recommended? The conclusion should contain information about the practical significance of the carried out research.
Reviewer 3 Report
The paper shows characterization of traditional Chinese sesame oil by different methods and discuss its errors in detail. The authors need to address the following concerns:
In section 2.2: the specifications of the HS-SMPE/GC-MS device (model, country of manufacture, etc.) should be mentioned.
In section 2.4: Was 5 grams of oil in a 25 ml bottle enough? I suppose more oil should have been poured into the sample bottle.
There are many factors that affect VOC emissions, have other interferences [factors other] than sesame levels been ruled out?
The effect of temperature and humidity on the measurements should be reported.
The significance of this paper [in Conclusions] is not expounded sufficiently. The authors need to highlight this paper's innovative contributions.
How many repetitions of each sample were used to measure Oxidation stability?
It is better to write the discussion section separately from the conclusion and expand it with previous research.
Some relative papers [references] regarding VOC sensors and e-nose may enrich the concepts and background of this work as references:
doi:10.1016/j.indcrop.2021.113355 ; doi:10.1007/s11694-020-00506-0
The conclusion section should be rewritten.
Round 2
Reviewer 2 Report
Manuscript has been revised according to the comments of the reviewer. It can be published.
Reviewer 3 Report
No comments